# The effects of prolonged single night session of videogaming on sleep and declarative memory

Miria Hartmann[1]☯*, Michael Alexander Pelzl[1]☯, Peter Herbert Kann[2], Ulrich Koehler[1‡], Manfred Betz[3], Olaf Hildebrandt[1‡], Werner Cassel[1]☯

1 Department of Pneumology, Intensive Care and Sleep Medicine, Philipps-University, Marburg, Germany, 2 Centre for Endocrinology, Diabetology & Osteology, Philipps-University, Marburg, Germany, 3 Faculty of Health Science, University of Applied Sciences, Gießen, Germany

☯ These authors contributed equally to this work.
‡ These authors also contributed equally to this work.
* miriahartmann@gmx.de

**Data Availability Statement:** All relevant data are within the manuscript and its Supporting Information files.

**Funding:** This study was funded by Loewenstein Medical GmbH. The funders had no role in study

## Abstract

Use of electronic media is widespread among adolescents. Many male adolescents spend a major part of their evenings playing video games. The increased exposure to artificial light as well as the exciting nature of this pastime is under suspicion to impair sleep. Sleep is considered to be important for memory consolidation, so there is also a potential risk for memory impairment due to video gaming. As learning and gaining knowledge is a very important part of adolescence, we decided to study the effects of prolonged video gaming on sleep and memory. The study was structured in a repeated measures design. Eighteen male participants played either the violent video game "Counter Strike: Global Offensive" or the board game "Monopoly" for five hours each on two Saturday nights. The game evenings were followed by sleep studies. Memory testing and vigilance evaluation was performed the next morning. During the course of the study, saliva samples were taken to determine melatonin and cortisol levels. The results of this crossover study showed slightly reduced sleep efficiency after "Counter Strike: Global Offensive" (-3.5%, p = .017) and impaired declarative memory recall (p = .005) compared to "Monopoly". Melatonin levels at bedtime were lower after "Counter Strike: Global Offensive" (p = .005), cortisol levels were elevated while playing the video game (p = .031). Negative effects on sleep were not strong but consistent with more wake after sleep onset (+12 min) and a higher arousal index after "Counter Strike: Global Offensive". We conclude that excessive video gaming in the evening can contribute to worsened sleep and impaired memory in male adolescents.

## Introduction

Usage of digital media is a part of the day-to-day life in industrialized countries [1]. Children's and adults' free time is largely spent using smartphones, computers and games consoles [2–4]. The innovation of electrical light and later the intense use of digital media is another general

design, data collection and analysis, decision to publish, or preparation of the manuscript.

**Competing interests:** The study was funded by the non-profit Löwenstein Foundation which had no role in study design, data collection, analysis and interpretation, decision to publish, or preparation of the manuscript. This does not alter our adherence to PLOS ONE policies on sharing data and materials.

behavioral change, humans have not been geared for by evolution and which has potential effects on sleep, wellbeing and learning performance [5]. Artificial light during the dark phase is known to impair melatonin secretion. This is especially true for light emitted by electronic screens which typically contains bluish light known to affect the internal clock and impair sleep quality [6]. Playing violent video games requires intense continuous screen observation. A recent large scale study [7] has shown, that 70% of the German male and 16% of the female adolescents play video games almost every day. 163 minutes of average gaming time per weekend day and 124 minutes per week day have been reported for male adolescents. In 2013 the American Psychiatric Association included Internet Gaming Disorder (IGD) in section III of the Diagnostic and Statistical Manual (DSM-5) [8]. The Worlds Health Organization also included Gaming Disorders in the actual version of the International Classification of Diseases (ICD-11) [9]. The measurement instruments and the consequences of Gaming Disorders are widely debated [10, 11]. Studies investigating video gaming (gaming time between 50 and 165 minutes) and sleep have shown variable effects on sleep and stress, the latter typically expressed as heart rate changes [12, 13]. Further studies showed inconclusive effects on sleep; most of them showed a modest effect [12, 14, 15]. Cortisol levels after videogaming were typically not higher than before videogaming [16] but until now cortisol levels during videogaming have not been measured. Sleep is closely associated with memory function and cerebral development of adolescents [17, 18]. Declarative learning and consolidation of knowledge is a key factor for successful school performance. Memory consolidation can be modulated by interfering activities between acquisition and reproduction. There is evidence for external disturbers like videogames or music during learning breaks [19] and also internal factors like cortisol levels [18, 20] interacting with memory consolidation. To our knowledge, there is no conclusive data about the effect of prolonged evening video gaming on memory function. Working in clinical sleep medicine we noted an increase in circadian sleep-wake-rhythm-disorders, type delayed sleep phase, over recent years, especially in male adolescents and young adults. Many of these patients reported excessive gaming time sometimes >8 hours per day, even on weekdays. We therefore decided to investigate effects of rather long gaming times. This study aims to collect data on the effects of prolonged time playing "Counter Strike: Global Offensive" (300 minutes) compared to playing "Monopoly" for the same duration in the evening before bed time. Measures included sleep (polysomnography), declarative memory performance, cortisol and melatonin levels, sleepiness and vigilance. Our main questions were:

- Does prolonged videogaming ("Counter Strike: Global Offensive") have a notable effect on sleep efficiency?

- Is there a difference in sleep-dependent post-sleep declarative memory recall between the "Counter Strike: Global Offensive" and "Monopoly" condition?

- In addition, we aimed to describe and compare:

- Cortisol levels during the study procedures; cortisol during "Counter Strike: Global Offensive" and during playing "Monopoly" was of special interest.

- Melatonin levels during the study procedures, especially at bed time.

- Vigilance in the morning after study nights.

- Differences in declarative memory performance directly after the "Counter Strike"/ "Monopoly" stimulation

## Methods

### Ethics

Ethic committee Marburg (Az.82/15) approved this study. All participants were informed in writing about the study and gave their written consent.

### Participants

Twenty male adolescents between 16 and 18 years of age were recruited via an article in the local newspaper. Regular school attendance, habitual daily video gaming and experience with violent video games were inclusion criteria. Preexisting sleep disorders, neurological disorders, cardiac diseases and regular intake of prescription medication led to exclusion. An introducing interview showed an average time in bed of the participants of 7,65±0,57 hours during the week and 9,02±1,26 hours at the weekend. Participants received 100 € remuneration for successful study completion. In case of participants aged under 18 the parents or guardians had to give their consent. Two participants discontinued the study because they felt it was too time-consuming, so eighteen participants (mean age 16.84 years) completed the study.

### Procedures

Investigations took place between September 2015 and February 2016. Each participant passed three study weekends. The first weekend served as habituation phase. Habitual sleepiness was measured (Epworth-Sleepiness-Scale [21]) at 6 PM on Saturday. Between 6 and 7 PM, the polysomnographic montage according to American Academy of Sleep Medicine (AASM) criteria [22] was attached. Participants were allowed to spend the evening in the sleep lab in whatever way they wished, which was typically either reading or watching TV. Between midnight and 7 AM on Sunday morning, polysomnography (PSG) was performed with Embla N7000 systems (Natus Medical Incorporated, San Francisco, CA.). During the following weekend, study procedures started at 3 PM on Saturday. After application of the PSG montage, all subjects underwent a memory test (Verbal Learn and Memory Test, VLMT [23]). The VLMT consists of two lists, each with fifteen unrelated words (list A and list B). There is a parallel form for repeated testing. List A is read out by the investigator in five consecutive learning trials, each followed by an immediate free recall by the participants. This is followed by one round of reading and immediate recall of the distractor list. After the "Counter Strike: Global Offensive" resp. "Monopoly" stimulation (pre-sleep) and on Sunday morning (post-sleep) the participants were asked to recall the words from list A. There was no exact time limit for the recall, but it usually lasted less than one minute until subjects stated that they recalled no further words. There was no feedback about the number of correct recalled words. Between 6 PM and 11 PM, four to five subjects jointly played either the board game Monopoly (HASBRO Deutschland GmbH, Dreieich) or the video game „Counter Strike: Global Offensive"(Valve, Bellevue, WA, recommended for players not younger than 16 years). Monopoly is a classic board game with the goal of earning the most money in comparison to the other players. "Counter Strike: Global Offensive" is one of the most popular video games and often played by adolescents. It´s a first-person shooter video game and the gamers are divided into a terrorist and a counter-terrorist team. Usually there are two different scenarios. The terrorists have to plant a bomb or defend the hostages while the counter-terrorists need to prevent planting or defuse the bomb or rescue the hostages. 50% of the participants played "Monopoly" on the second weekend and the video game on the third weekend, the other half played in reverse sequence (Sequence 1: Counter Strike/Monopoly; Sequence 2: Monopoly/Counter Strike). The two parallel forms of the VLMT were used for the two study weekends and subjects were informed about the

planned recalls on both weekends. Unfortunately, the two drop-outs were both in group sequence 2 but in all it was nearly balanced (Sequence 1: ten data sets, Sequence 2: eight data sets). The participants were offered pizza and snacks in the evening. When playing a game for five hours was too exhausting for a participant, he was motivated to continue so none of the adolescents had to stop playing during the measurements. During video gaming, participants were subjected to 45–55 lux light intensity at eye level from the PC screens. Ambient light intensity during the board game was around 20 lux at eye-level. After playing games, participants got ready for the night and PSG recordings were started (lights out) around midnight in single room sleep labs. Lights on was at 7 AM. Subjects filled in the Stanford Sleepiness Scale [24] to evaluate current sleepiness 30 minutes after lights on. Subsequently they underwent the pupillographic sleepiness test (PST) [25] and the vigilance test VigiMar [26]. For the 11-minutes PST, pupil diameter variations indicate vigilance, whereas mean reaction time over a monotonous 90 minutes four choice reaction time test is used as vigilance measure of the VigiMar. In order to determine cortisol and melatonin concentration, five saliva samples were taken at 5.45 PM, during a short gaming break at 9.07 PM, after gaming at 11.37 PM, during nocturnal rest at about 2.10 AM and after lights on at 7.28 AM. To minimize sleep disruption, the fourth nocturnal saliva sample was taken only when subjects were currently not in REM or slow wave sleep, so sample times were more variable than for the other saliva samples (2.10 AM ± 19 minutes). While collecting the nocturnal sample the ceiling light was not switched on.

Fig 1 gives a temporal overview for the study weekends two and three.

## Data analysis

Polysomnographic recordings were analyzed visually according to standard criteria [22] by trained and certified (German Sleep Society) sleep technicians. Results of memory testing (VLMT) are determined by adding up the number of correct words recalled. The maximum attainable score is 15 (all words recalled). Pupil diameter variations are expressed as pupillary unrest index (PUI). High PUI values are computed by the PST system to indicate low vigilance. The VigiMar test computes mean reaction time over the test. Slow reaction time indicates low vigilance. Melatonin was measured with the ELISA-SLV-4779 enzyme immunoassay salivary kit. Active free cortisol levels were determined with the salivary kit HS ELISA-SLV-4635 (both kits DRG International Inc).

## Statistical analysis

Statistical Analysis was done with SPSS version 22 (IBM corporation). All variables were tested for normality by Kolmogorov-Smirnov one sample testing. Most variables showed deviations

**Fig 1. Temporal overview of measurements.** VLMT- Verbal Learn and Memory Test.

from a normal distribution. We therefore decided to use robust non-parametric methods for all statistical testing. As most studies in this field report results that are expressed as means and standard deviations, we decided to use these measures along with median and quartiles for descriptive purposes in order to facilitate comparisons of our results to the literature. All tables therefore contain mean and standard deviation as well as median and 1$^{st}$ and 3$^{rd}$ quartile. Our study addresses two non-independent main issues (issue 1: sleep efficiency, issue 2: declarative memory). Bonferroni adjustment of the $\alpha$-error was performed, which resulted in p = .025 as critical value for statistical significance. The comparison for sleep efficiency and memory between "Counter Strike: Global Offensive" and "Monopoly" condition was done by Wilcoxon signed rank test for repeated measures. The comparison for secondary variables (cortisol, melatonin, vigilance, other sleep variables) was also done by Wilcoxon signed rank test; the resulting p-values were not used for generalization of these results but as a measure of effect robustness instead. We calculated r-values as measures of effect sizes for all reported statistically significant tests according to Cohen who recommended Pearson r-values of .10, .30 and .50 to demarcate small, medium, and large effects [27, 28].

## Results

### Main questions (adjusted significance level p $<$ .025)

For the night following the evening of "Counter Strike: Global Offensive" (CS) sleep efficiency was significantly reduced in comparison to the night following the evening of "Monopoly" (MP) (CS = 88.56% ± 7.14%; MP = 92.08% ± 2.8%; p = .017; r = .40). The recall of the fifteen words presented in the VLMT as a measurement of conscious declarative learning was worse on the morning after "Counter Strike" when participants recalled 9.56 ± 3.62 words compared to 11.83 ± 2.36 words after the "Monopoly" condition (post-sleep; p = .005; r = .46).

### Additional sleep variables

The slightly lower sleep efficiency after "Counter Strike" seems to be equally due to longer sleep latency (CS = 10.93 ± 10.42 min; MP = 7.43 ± 4.91 min) and longer wake after sleep onset after "Counter Strike" (CS = 38.36 ± 24.58 min; MP = 26.25 ± 10.52 min).

Post videogaming sleep contained more light sleep stage N1 (CS = 32.17 ± 19.83 min; MP = 28.69 ± 15.73 min; p = .0497) and less sleep stage N2 (CS = 171.33 ± 30.53 min; MP = 188.24 ± 25.1 min; p = .031). There was also a small difference in the number of arousals per hour of sleep (CS = 8.85 ± 3.97; MP = 7.12 ± 2.41; p = .035).

Table 1 summarizes sleep results.

### Free recall directly after stimulation

As we expected that interference of the more exciting videogame "Counter Strike: Global Offensive" may contribute to performance differences in declarative learning, we included a quick free recall of the previously learned 15 words directly after the stimulation on the same evening without correcting words or repeating the list. This already showed a tendency towards a worse performance in the "Counter Strike: Global Offensive" condition (Correct words pre-sleep CS = 10.56 ± 3.07; MP = 12.17 ± 2.01; p = .016; r = .40).

### Vigilance and sleepiness

Self-reported acute sleepiness after the study nights (7.30 AM) did not differ between the "Counter Strike: Global Offensive" night and the "Monopoly" night (SSS-Score CS = 3.47 ± 1.00; MP = 3.33 ± 1.24; p = .816). The same applies for vigilance according to

**Table 1. Sleep results.**

| | Counter Strike: Global Offensive | Monopoly | p-value | r-value (if statistically significant) |
|---|---|---|---|---|
| Sleep Efficiency (%) | 88.56 ± 7.14 | 92.08 ± 2.87 | **.017** | .40 |
| | 90.2, 85.05; 94.4 | 93.3, 89.6; 94.13 | | |
| Sleep Latency (min) | 10.93 ± 10.42 | 7.43 ± 4.91 | .193 | |
| | 7.49, 2.88; 15.75 | 5.75, 3.88; 11.88 | | |
| Wake after Sleep Onset (min) | 38.36 ± 24.58 | 26.25 ± 10.52 | .064 | |
| | 31.68, 21; 53.25 | 24, 19.88; 33.82 | | |
| N1-Duration (min) | 32.17 ± 19.83 | 28.69 ± 15.73 | **.0497** | .33 |
| | 33.25, 13.88; 49.25 | 28.5, 11.88; 40.25 | | |
| N2-Duration (min) | 171.33 ± 30.53 | 188.24 ± 25.1 | **.031** | .36 |
| | 171.25, 149.13;19.13 | 189, 174; 214.13 | | |
| N3-Duration (min) | 105.72 ± 35.94 | 101.72 ± 29.7 | .570 | |
| | 100.75, 79.25; 123.25 | 94, 81.25; 118.75 | | |
| REM-Duration (min) | 72.43 ± 17.29 | 78.86 ± 16.35 | .127 | |
| | 72.87, 56.25; 84 | 82.5, 64.63; 92.63 | | |
| Arousal- Index | 8.85 ± 3.97 | 7.12 ± 2.41 | **.035** | .35 |
| | 7.99, 5.49; 11.9 | 7.62, 4.81; 9.36 | | |

Sleep variables for "Counter Strike: Global Offensive" and "Monopoly" (mean ± SD; median; 1st quartile; 3rd quartile; relevant results marked in bold).

pupillography (PUI: CS = 5.85 ± 2.49; MP = 5.80 ± 2.10; p = .914). VigiMar results were also very similar for both conditions: Mean reaction time did not differ in a relevant way (CS = 2.10 ± 7.06 sec; MP = 1.94 ± 2.05 sec; p = .647). The results of the three tests are presented in Table 2.

## Melatonin

Post gaming melatonin levels between 11 PM and 12 PM close to lights out were clearly lower after "Counter Strike: Global Offensive" compared to "Monopoly" (CS = 5.74 ± 5.33 pg/ml; MP = 12.30 ± 9.80 pg/ml; p = .005; r = .47). No statistically relevant differences between both conditions were observed for other acquisition times. The pattern of melatonin level illustrated in Fig 2 indicates lower melatonin levels at lights out and nominally (but not significantly) higher values at lights on after "Counter Strike: Global Offensive". All melatonin results are shown in Table 3.

**Table 2. Vigilance results.**

| | Counter Strike: Global Offensive | Monopoly | p-value |
|---|---|---|---|
| SSS | 3.47 ± 1.00 | 3.33 ± 1.24 | .816 |
| | 4.00, 3.00; 4.00 | 3.00, 2.75; 4.00 | |
| PUI (PST, mm/min) | 5.85 ± 2.49 | 5.80 ± 2.10 | .879 |
| | 5.59, 3.73; 8.15 | 5.94, 4.07; 7.17 | |
| MRT (Vigimar, sec) | 2.10 ± 7.06 | 1.94 ± 2.05 | .453 |
| | 1.19, 0.80; 3.05 | 1.27, 0.91; 1.72 | |

Vigilance results for "Counter Strike: Global Offensive" and "Monopoly". SSS-Stanford Sleepiness Scale, PUI: Pupillary-Unrest-Index, MRT: Mean Reaction Time (mean ± SD; median; 1st quartile; 3rd quartile).

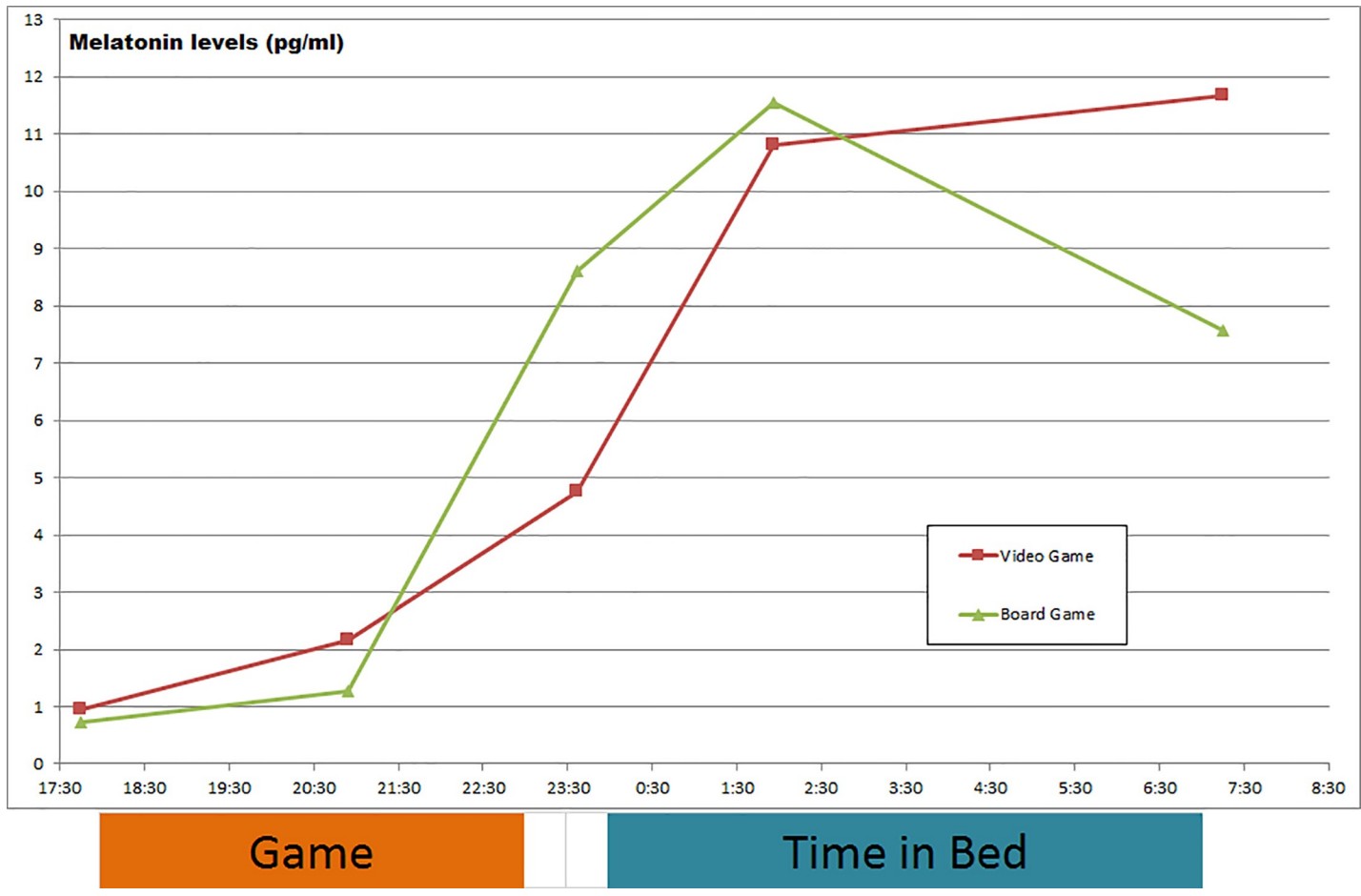

**Fig 2. Saliva melatonin levels (pg/ml) for the "Counter Strike: Global Offensive" and "Monopoly" conditions.**

## Cortisol

Cortisol values obtained while playing games (short break at 9.07 PM) showed significantly higher values during "Counter Strike" (CS = .72 ± .26 ng/ml; MP = .55 ± .21 ng/ml; p = .031;

**Table 3. Melatonin results.**

|  | Counter Strike: Global Offensive | Monopoly | p-value | r-value (if statistically significant) |
|---|---|---|---|---|
| **5.45 PM** | 1.62 ± 2.29 | 0.92 ± 1.01 | .278 |  |
|  | 0.95, 0.38; 1,89 | 0.71, 0.00; 1.29 |  |  |
| **9.07 PM** | 2.60 ± 2.96 | 2.93 ± 3.84 | .836 |  |
|  | 2.17, 0.00; 4.09 | 1.27, 0.62; 4.16 |  |  |
| **11.37 PM** | 5.74 ± 5.33 | 12.30 ± 9.80 | **.005** | .47 |
|  | 4.75, 0.81; 10.49 | 8.62, 4.12; 14.37 |  |  |
| **2.10 AM** | 11.50 ± 7.45 | 12.96 ± 7.47 | .472 |  |
|  | 10.80, 8.48; 13.85 | 11.54, 8.98; 13.28 |  |  |
| **7.28 AM** | 13.31 ± 9.99 | 11.28 ± 10.43 | .157 |  |
|  | 11.66, 5.05; 15.74 | 7.57, 4.24; 13.35 |  |  |

Saliva melatonin levels for "Counter Strike: Global Offensive" and "Monopoly" (pg/ml; mean ± SD; median; 1st quartile; 3rd quartile; relevant results marked in bold).

**Table 4. Cortisol results.**

| | Counter Strike: Global Offensive | Monopoly | p-value | r-value (if statistically significant) |
|---|---|---|---|---|
| **5.45 PM** | 1.33 ± 0.89 | 1.45 ± 0.69 | .601 | |
| | 1.15, 0.85; 1.35 | 1.28, 0.87; 1.97 | | |
| **9.07 PM** | 0.72 ± 0.26 | 0.55 ± 0.21 | **.031** | .36 |
| | 0.67, 0.55; 0.83 | 0.48, 0.44; 0.73 | | |
| **11.37 PM** | 0.65 ± 0.68 | 0.54 ± 0.29 | .879 | |
| | 0.50, 0.35; 0.60 | 0.54, 0.34; 0.67 | | |
| **2.10 AM** | 0.83 ± 0.40 | 0.58 ± 0.27 | .081 | |
| | 0.74, 0.56; 0.90 | 0.59, 0.41; 0.78 | | |
| **7.28 AM** | 4.49 ± 1.83 | 4.48 ± 1.98 | .948 | |
| | 3.92, 3.29; 5.62 | 3.84, 3.17; 5.82 | | |

Saliva cortisol levels for "Counter Strike: Global Offensive" and "Monopoly" (ng/ml; mean ± SD; median; $1^{st}$ quartile; $3^{rd}$ quartile; relevant results marked in bold).

r = .36). This difference is no longer present after gaming. Cortisol values during the night and in the morning are similar for both gaming conditions. Neither the pre- nor the post-gaming cortisol level show any difference. Table 4 and Fig 3 demonstrate all cortisol values obtained.

## Discussion

### Discussion of methods

Most studies about video gaming and sleep had relatively low sample sizes. Higuchi et al. had seven PSG recordings, Weaver et al. nine and Dworak et al. ten recordings [12, 13, 29]. King et al. had objective sleep data from seventeen subjects which is close to our sample size of eighteen participants [14]. Compared to the existing studies, our sample size is rather high. Furthermore, taking night to night variability in sleep studies into account, even larger sample sizes are desirable. As videogaming is more prevalent in male adolescents we decided to include only male subjects [7]. This homogeneity leads to a better relation between effect-variability and between-subjects-variability. Of course, this means that our results should only be generalized for male subjects. We chose long gaming durations of 300 minutes for each condition which is long compared to other studies (60 min Dworak et al. [29]; 50 min Weaver et al. [12]; 165 min Higuchi et al. [13]. However, on average, male German adolescents play videogames for more than 160 minutes on weekend days [7]. For weekdays, 5.5% of male adolescents report playing videogames for 300 minutes or more and on weekends, this proportion rises to 15.9%. In addition, total daily usage time of electronic media for German adolescents is greater than five hours per day [30]. So 300 minutes gaming time seems long, but is definitely not beyond daily routine experience of male adolescents. Accordingly, not a single study participant found 5 hours of "Counter Strike" unusual or too long, while several complained about five hours of playing "Monopoly" being too long. Another reason for the long gaming time was our intent to maximize potential effects in our small sample. This experimental design compares an exciting computer game with the rather boring board game "Monopoly". For a better comparability a more fast-paced board game could have been used. Many sleep studies use a standardized time in bed of eight hours. We chose a shorter time in bed of seven hours. A large scale study with 5275 German adolescent subjects [31] showed an average 6 h 47 min reported weekday time in bed, so our time in bed is close to the time spent in bed in real life. Due to a tight schedule in the sleep laboratory and the duration of playing five hours the games in teams, it was not possible to adjust the bed/wake times to the habitual sleep times.

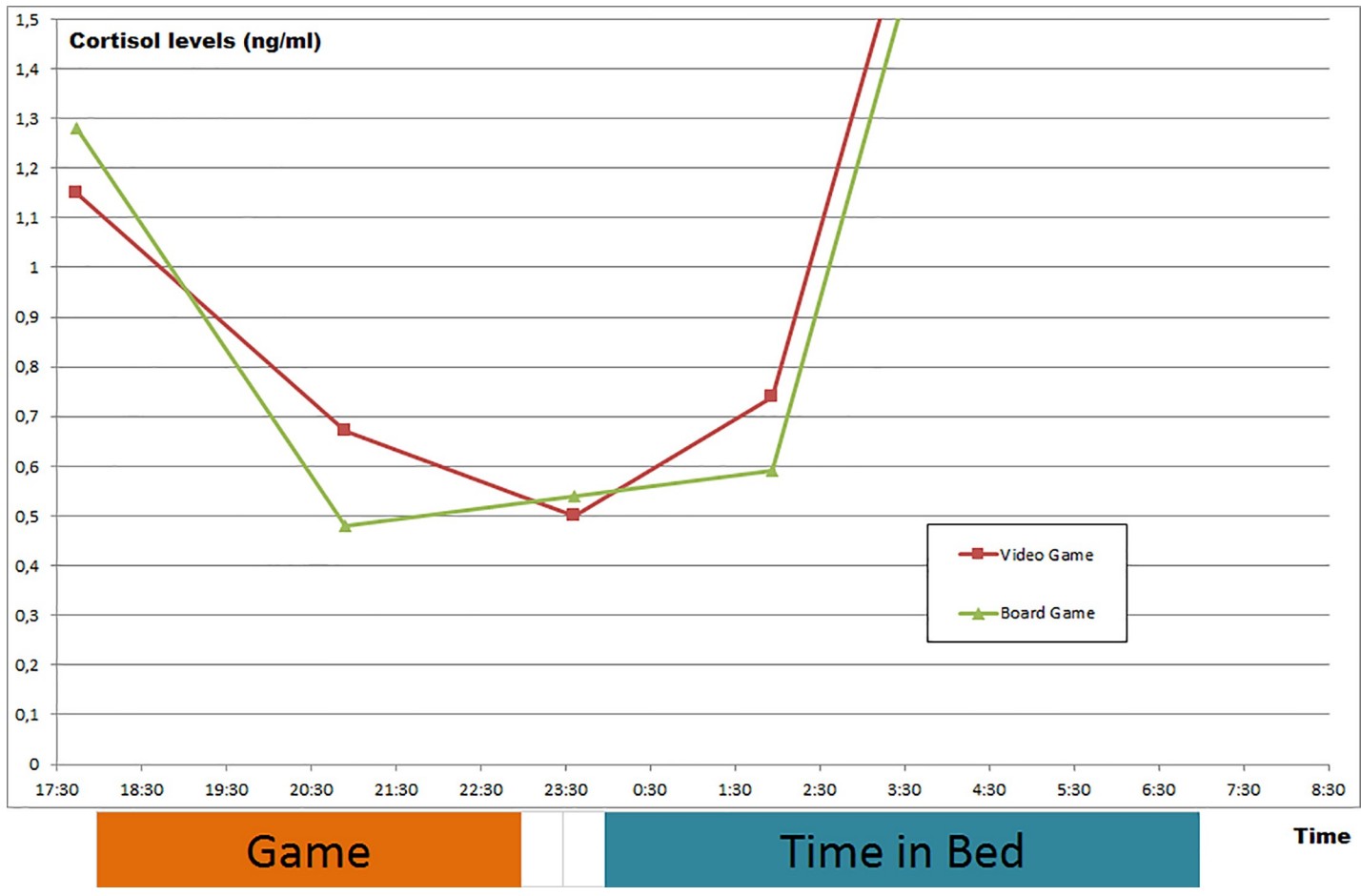

**Fig 3. Saliva cortisol levels (ng/ml) for the "Counter Strike: Global Offensive" and "Monopoly" conditions.**

In comparison to the given time in bed of our participants (7,65±0,57 hours during the week and 9,02±1,26 hours at the weekend), seven hours are quite short. Different bed times could influence the outcome of the sleep measurements.

## Discussion of results

We found a negative effect of playing "Counter Strike" on both our main variables. Sleep efficiency was slightly but significantly reduced by 3.5% after prolonged playing "Counter Strike" compared to the same duration of playing Monopoly. This translates to fifteen more minutes spent in wakefulness during time in bed. King et al. also found lowered sleep efficiency after prolonged videogaming, whereas other authors found no effect on sleep efficiency [13, 14, 29]. Recall after declarative learning was reduced on the morning after playing "Counter Strike" compared to the "Monopoly" condition. Bad sleep [32] as well as interference between learning and retention [17, 33] can impair memory function. In post hoc analysis of the remembered words directly after the stimulation (pre-sleep), we already found a tendency of the declarative memory performance being worse after playing "Counter Strike" (p = .031). This difference gets more robust after the consolidation during the night at the next morning free recall (p = .005). All statistically significant results fall into the medium effect size range (r = .30 - .50) according to Cohen [27,28]. With our sample size and study design it is not feasible

to measure the exact contributions of the two expected main effects on declarative learning: interference and consolidation. Most likely both contribute to memory performance being worse in the morning after playing "Counter Strike", the parameter we considered most relevant as it shows what our subjects actually remembered. In recent papers about memory encoding it was stated that a rise in cortisol levels can facilitate recall of learned information, but only if the hormone levels are high in pre-learning stage [20] and low during encoding and consolidation [18]. In comparison of "Counter Strike" and "Monopoly" we saw significantly rising cortisol levels after the learning phase (during the game) and again elevated cortisol levels in the sleep stage (not significant). This could have led to more interference of the impressions during the game with the previously learned words, as the new impressions were likely "tagged" as more important because of higher stress levels [20], and less effective consolidation in sleep [18]. Following this hypothesis, maybe it would have made more sense to play "Counter Strike" before learning and go to sleep after learning. This could be an interesting subject for further studies. Relating to additional results, more time spent in N1 and a slightly higher arousal-index can be viewed as indicators for lighter sleep after "Counter Strike". In line with these sleep results Melatonin values after gaming and before bedtime were lower in the videogame condition. Ivarsson et al. did not find increased arousals after violent videogames compared to non-violent games, but they described more awakenings in frequently videogaming adolescents compared to adolescents who scarcely used videogames [34, 35]. Higuchi et al. found lowered REM sleep after videogaming [13]. While we also observed seven minutes less of REM sleep after "Counter Strike", this difference failed to reach statistical significance in our study. This might partially be due to our short time in bed of seven hours which curtailed sleep at the end of the resting period, the time when REM is more probable than in the first half of the night. Prolonged evening videogaming of "Counter Strike: Global Offensive" seems to impair melatonin secretion as indicated by significantly lower melatonin levels at bedtime. Higher light exposure during "Counter Strike" (50 lux vs. 20 lux during "Monopoly" what is quite dim) with constant observation of a bright computer monitor can cause this effect. Several studies have shown that light from a computer monitor is sufficient to reduce and phase-delay melatonin secretion [5, 36]. Accordingly, our data indicates lower melatonin levels at bedtime and higher absolute melatonin levels in the morning (not significant) after "Counter Strike", while maximum levels at 2 AM are similar for both conditions. Excitation from video gaming can contribute to this effect. Excitement and tension can cause pupil dilatation via sympathetic arousal [37], which increases sensitivity to light and reduces melatonin secretion [38]. We are aware of one other study which addresses melatonin levels and videogaming. Higuchi et al. demonstrated significantly reduced melatonin levels after playing exciting videogames in front of bright computer screens [39]. Solving non-exciting tasks presented on screens with similar brightness did not impair melatonin secretion. These results also indicate an interaction between light and excitement. Eyes with wide pupils are more sensitive to light which increases the risk for reduced melatonin production [38]. Several studies found no increases in stress hormone cortisol levels for pre-post videogaming comparisons [16, 34]. These findings are in line with our results for the pre-post comparison and frequently lead to the conclusion that even violent videogaming compared to no gaming at all does not increase cortisol secretion [34]. In contrast to existing studies we took cortisol samples in a short break in the middle of the videogaming period. In agreement with the well-known circadian evening decline in cortisol [40], we found lower cortisol levels during both gaming periods than before gaming started. But mid-gaming cortisol levels during "Counter Strike" were significantly higher than those observed during "Monopoly". So, our results indicate slightly higher levels of the stress hormone cortisol during the exciting video game "Counter Strike". In agreement with hints for slightly impaired sleep after the violent videogame "Counter

Strike" we found a trend towards slightly higher cortisol levels during the nocturnal resting period. Sleepiness and vigilance testing results did not differ for both gaming conditions. This is not surprising as the observed effects on sleep after one evening of prolonged playing the violent videogame "Counter Strike" were small and not clinically meaningful. Nevertheless, there is data indicating that adolescents with habitual high levels of electronic media use are sleepier [41]. The lack of effect in this area is probably due to the single stimulus used in this study. Repeated exposure to prolonged gaming might yield clearer results.

## Conclusion

Prolonged playing the video game "Counter Strike: Global Offensive" seems to impair sleep and declarative memory. We consider these medium sized effects remarkable as they occur after different leisure activities and not after formal interventions directly aimed at sleep or memory. Future studies should aim to disentangle the effects of nocturnal light and inherent excitement and engagement associated to violent video gaming from each other. It may also be interesting for future studies to find out to which extent observed declarative memory performance differences are caused by interference after learning in contrast to consolidation during sleep. In addition to that an exciting video game as "Counter Strike: Global Offensive" is compared to the rather boring board game "Monopoly". Future studies could compare an exciting video game to an also more exciting board game.

## Supporting information

**S1 File. Data availability.**
(XLSX)

## Author Contributions

**Conceptualization:** Peter Herbert Kann, Ulrich Koehler, Manfred Betz, Olaf Hildebrandt, Werner Cassel.

**Data curation:** Miria Hartmann, Michael Alexander Pelzl, Olaf Hildebrandt, Werner Cassel.

**Formal analysis:** Miria Hartmann, Michael Alexander Pelzl, Werner Cassel.

**Funding acquisition:** Ulrich Koehler, Olaf Hildebrandt.

**Investigation:** Miria Hartmann, Michael Alexander Pelzl, Ulrich Koehler.

**Methodology:** Ulrich Koehler, Manfred Betz, Olaf Hildebrandt, Werner Cassel.

**Project administration:** Ulrich Koehler, Olaf Hildebrandt.

**Resources:** Peter Herbert Kann, Ulrich Koehler, Olaf Hildebrandt.

**Supervision:** Ulrich Koehler, Olaf Hildebrandt, Werner Cassel.

**Validation:** Michael Alexander Pelzl, Manfred Betz, Olaf Hildebrandt, Werner Cassel.

**Visualization:** Olaf Hildebrandt.

**Writing – original draft:** Miria Hartmann, Michael Alexander Pelzl, Ulrich Koehler, Werner Cassel.

**Writing – review & editing:** Miria Hartmann, Michael Alexander Pelzl, Peter Herbert Kann, Ulrich Koehler, Manfred Betz, Olaf Hildebrandt, Werner Cassel.

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
