## [Decision Letter · Decision Letter 0]

24 Jul 2019

PONE-D-19-17611

The effects of prolonged videogaming on sleep and declarative memory

PLOS ONE

Dear Mrs. Hartmann,

Thank you for submitting your manuscript to PLOS ONE. After careful consideration, we feel that it has merit but does not fully meet PLOS ONE’s publication criteria as it currently stands. Therefore, we invite you to submit a revised version of the manuscript that addresses the points raised during the review process.

I agree that the manuscript presents an interesting research and deserves consideration, given that adequate revision will be provided by Authors basing on Reviewers' concerns. 

In particular, I agree that the manuscript's tone should be softened at times, basing on what it has actually been done. Possibly the main limitation of the study is the usage of Monopoli, while Authors could easily find some fast-paced, stressful, combat-themed board game allowing for more proper comparison. I suggest Authors to (1) add a more specific limitations section with this and other limitations identified by Reviewers explicitly stated, and (2), when referring to the experiment (text, tables, figures included), to not report the comparison between "video gaming and board gaming", but between "counter strike and monopoli", as these products are not necessarily representative of all the extremely rich gaming scenario. 

About this: the two games used should be described in more detail, given that they are the only experimental operators.

Future research section could be improved as well, taking into account that comparison beetween different board/video games could potentially lead to opposite results. 

We would appreciate receiving your revised manuscript by Sep 07 2019 11:59PM. To enhance the reproducibility of your results, we recommend that if applicable you deposit your laboratory protocols in protocols.io, where a protocol can be assigned its own identifier (DOI) such that it can be cited independently in the future. For instructions see: http://journals.plos.org/plosone/s/submission-guidelines#loc-laboratory-protocols

We look forward to receiving your revised manuscript.

Kind regards,

Stefano Triberti, Ph.D.

Academic Editor

PLOS ONE

Journal Requirements:

2) Please state in your methods section whether you obtained consent from parents or guardians of the minors (participants aged under 18) included in the study or whether the research ethics committee or IRB approved the lack of parent or guardian consent.

3) We note in your conclusion you indicate that you obtained 'surprisingly large' effects on video gaming on declarative memory. However, no effect sizes have been reported in the Results section. Please correct and/or clarify how you determined your effects were large.

4)  Thank you for stating the following in the Competing Interests section:

[The authors have declared that no competing interests exist.].

We note that you received funding from a commercial source: Loewenstein Medical GmbH.

5) We note that you have stated that you will provide repository information for your data at acceptance. Should your manuscript be accepted for publication, we will hold it until you provide the relevant accession numbers or DOIs necessary to access your data. If you wish to make changes to your Data Availability statement, please describe these changes in your cover letter and we will update your Data Availability statement to reflect the information you provide.

Reviewers' comments:

Reviewer's Responses to Questions

**Comments to the Author**

1. Is the manuscript technically sound, and do the data support the conclusions?

Reviewer #1: Yes

Reviewer #2: Yes

2. Has the statistical analysis been performed appropriately and rigorously? 

Reviewer #1: Yes

Reviewer #2: Yes

3. Have the authors made all data underlying the findings in their manuscript fully available?

Reviewer #1: Yes

Reviewer #2: Yes

4. Is the manuscript presented in an intelligible fashion and written in standard English?

Reviewer #1: Yes

Reviewer #2: Yes

5. Review Comments to the Author

Reviewer #1: Dear authors,The paper needs further investigation.

1 - You used a violent video game (Global strike - Global Offensive) and a board game (Monopoly). Why didn't you use other kinds of video games? Video game is very activating (see Kovess-Masfety et al (2016). Is time spent playing video games associated with mental health, cognitive and social skills in young children?). While the board game is less activating (the arousal of the BN is lower) and consequently you have received more complaints about 5 hs of playing a board game being too long. It is considered a boring activity, obviously sleep arises first.

2 - Why did the subjects participate in both experimental conditions? In this way it was possible to create an expectation effect.

3 – In “Procedure” you say that “50% of the participants played the board game on the second weekend and the video game in the third weekend, the other half played in reverse sequence”. Have you analyzed if the different sequences (VN-BN and BN-VN) gives different effects on variables? Compare the results of variable in the two groups that followed different sequences.

4 - It is true that 300 minutes of playing video games are many, but the exposure is only one. Specify in the paper that the effects occur following a short-term exposure. The title could be changed to "The effects of prolonged single night session of videogaming on sleep and declarative memory"

5 – In the first phrase of the paragraph “Discussion of Results”, word “detrimental” is too strong. Replace with a more suitable term.

6 – Which Correct Words value is reported in the paper? Pre- or Post- sleep? This is unclear.

7 - Why didn't you use a non-verbal declarative memory test? For example a visual declarative memory test. Other information could emerge (see Peracchia, S., & Curcio, G. (2018). Exposure to video games: effects on sleep and on post-sleep cognitive abilities. A sistematic review of experimental evidences)

8 - Update the literature.

Reviewer #2: The experiment presented by Hartmann et al. investigates the effect of prolonged evening usage of videogames on declarative memory and sleep. They find that sleep efficiency and declarative memory performance were both significantly reduced and speculate that these findings may be due to increased interference and impaired overnight consolidation in the videogame condition, although they are unable to fully disentangle the two.

Since increased screen time and gaming are both increasingly prevalent in modern society, this is a relevant research question and the results are suitable for publication in PLOS One. My concerns rest primarily with the lack of detail in the methodology, specifically:

1. How was the VLMT administered? The text states that the list was memorized through five repetitions, but do not state whether words were presented one at a time, all together, etc., how much time participants were given to study the list, how much time passed in between repetitions, whether it was a computerized task, etc. Nor do the authors mention how much time participants were given for the free recall of the word list. It is not clear what is meant by “the recall was preannounced in each repetition” –does this mean participants were explicitly instructed to memorize the words?

2. For the 5 hours of gaming, were participants gaming continuously without breaks (other than the scheduled break at 8:55 to collect saliva)? It is also unclear whether the PCs were the sole source of light while playing video games or if there was additional ambient room lighting. Was the 20 lux of ambient lighting during the board game also measured at eye level? I am also curious how 20 lux was selected for the ambient lighting during the board game, as it is quite dim compared to ordinary room lighting. Finally, what lux was the room during the sleep period?

3. Were habitual bed/wake times taken into account during screening? Especially because sleep is one of the primary outcome measures, it is important to note whether the lab-scheduled sleep opportunities are at similar times to the participants’ habitual sleep times. If participants are attempting to sleep at times very different from their normal time, their sleep may be impacted independent of gaming condition.

4. How was the timing of the saliva samples chosen? Also, it would be helpful to report the average time and standard deviation of the fourth nocturnal saliva sample, so that readers are aware of the variability of this sample time.

5. Why was the VigiMar test used to assess vigilance instead of the more commonly used Psychomotor Vigilance Test (PVT)? And given that the VigiMar is a very long test, were there any differences in time-on-task effects in the two conditions?

6. The authors report significantly lower melatonin levels after video gaming compared to board games (pg 10) and state that this “indicates a phase delay of melatonin secretion”. While this clearly indicates melatonin suppression, and it is possible that evening video games could have caused a phase delay, the current experimental design does not allow them to assess whether there has actually been a phase shift. Similar on pg 14, line 360, the data show melatonin suppression but not necessarily a phase delay.

7. Although the authors note that participants were scheduled to 7 hours time in bed, which is close to the average for German adolescents, they also report on page 5 that their participants reported sleeping on average 7.65 hrs in bed during the week and 9.02 hrs on the weekend. Thus, it is likely that they were either somewhat sleep restricting the participants, or at the very least scheduling wake times significantly different from the participants’ habitual wake. This should be mentioned in the discussion of scheduled time in bed.

8. Minor typographical errors in the text:

a. Use consistent tense throughout the manuscript. E.g. on pg 2, line 28, change “does” to “did”; pg 14 line 355, change “seems” to “seemed”, etc.

b. Pg 5 in the ethics statement, it is not clear what “All participants were tired and informed…” is intended to say.

c. Pg 6, line 172 “brake” should be changed to “break”

d. Pg 15, line 393 “extend” should be changed to “extent”

6. PLOS authors have the option to publish the peer review history of their article (what does this mean?). If published, this will include your full peer review and any attached files.

Reviewer #1: No

Reviewer #2: No

---

## [Author Response · Author response to Decision Letter 0]

6 Sep 2019

Reviewer 1:

1) You used a violent video game (Global strike - Global Offensive) and a board game (Monopoly). Why didn't you use other kinds of video games? Video game is very activating (see Kovess-Masfety et al (2016). Is time spent playing video games associated with mental health, cognitive and social skills in young children?). While the board game is less activating (the arousal of the BN is lower) and consequently you have received more complaints about 5 hs of playing a board game being too long. It is considered a boring activity, obviously sleep arises first.

Response: "Counter Strike: Global Offensive" is one of the most popular video games among adolescents and all of our participants had at least some experience with the game. We used this violent video game because of its high topicality, the possibility of team playing including fighting against each other. In our small sample, we wanted to maximize potential effects by playing a violent game for a very long time (five hours). Since 2018, after our study measurements, the video game "Fortnite" has become the most popular video game among 12 – 19 years old German adolescents (Medienpädagogischer Forschungsverbund Südwest. JIM 2018 Jugend, Information, (Multi-)Media: Basisstudie zum Medienumgang 12- bis 19-Jähriger in Deutschland 2018). Nowadays this game could be an alternative video game condition. 

In line with the Academic Editor Mr. Triberti we added a paragraph about the comparison of the rather boring “Monopoly” and the more exciting computer game (l. 320-322; l. 412-415). In general, we consciously intermingled the aspect “exciting” with video gaming and “less exciting” with board gaming because we wanted to maximize effects. The drawback that inevitably goes along with that decision has been emphasized in the new version of the manuscript. Another not so scientific reason for choosing monopoly were the reports of the more senior co-authors (Koehler, Cassel) that they had spent considerable time with playing Monopoly as adolescents.

2) Why did the subjects participate in both experimental conditions? In this way it was possible to create an expectation effect.

Response: The first weekend was just a habituation phase with no gaming conditions. The adolescents participated in a reverse sequence model so 50 % played "Counter Strike: Global Offensive" at the second weekend and "Monopoly" at the third weekend. The other 50 % played "Monopoly" first and then "Counter Strike: Global Offensive". Interindividual sleep can vary widely. With our small sample size of 20 (including two drop outs) participants we wanted to prevent a high interindividual variation. While independent samples would have avoided the expectation effect, they would have introduced more interindividual variations requiring larger sample sizes which would have been very difficult to achieve.

3) In “Procedure” you say that “50% of the participants played the board game on the second weekend and the video game in the third weekend, the other half played in reverse sequence”. Have you analyzed if the different sequences (VN-BN and BN-VN) gives different effects on variables? Compare the results of variable in the two groups that followed different sequences.

Response: Thank you for this suggestion. We have compared the results in the two different groups. Sequence 1: Counter Strike - Monopoly; Sequence 2: Monopoly - Counter Strike. Unfortunately, the two drop outs of the study were both in the sequence 2 group, so there are just eight data sets in sequence 2 and ten data sets in sequence1 (l. 168-172 in the manuscript).

There was a sequence effect in the following variables (CS=VN, MP=BN):

Sleep variables:

N1-Duration MP: Seq 1 = 35,35 ±13,09; Seq 2 = 20,38 ± 15,42; p = .034

N3-Duration CS: Seq 1 = 90,85 ± 32,38; Seq 2 = 124,32 ± 32,82; p = .027

Pupillographic sleepiness test: 

PUI CS: Seq 1: 4,45 ± 1,97; Seq 2: 7,59 ± 1,95; p = .004

VigiMar:

MRT CS: Seq 1: 1,17 ± 0,62; Seq 2: 3,24 ± 2,34; p = .027

Cortisol:

Cortisol 11.37 PM MP: Seq 1: 0,69 ± 0,25; Seq 2: 0,35 ± 0,20; p = .003

Cortisol 02.10 AM MP: Seq 1 0,69 ± 0,14; Seq 2: 0,44 ± 0,33; p = .016

 (Extract of our excel list. Please have a look at our document "Response to the reviewers")

To sum up, we can identify the following points:

1. Those who played Counter Strike in the first night 

- had less deep sleep after the videogame but more light sleep after Monopoly

- were more alert on the next morning after Counter Strike during the VigiMar and PST

Despite of the first habituation night, the combination of Counter Strike and the first night with measurements could have been more exciting than Monopoly and measurements. 

2. Those who played Counter Strike in the first night 

- had higher cortisol levels directly after and in the night after Monopoly

Perhaps the adolescents were more annoyed of the board game. They have already played the more entertaining computer game on the other weekend. In regards to the group structure of four or five adolescents they were glad and excited that the measurements come to an end so they pushed each other for the last time. 

Admittedly, these findings and interpretations are rather vague so we did not include them into the manuscript. We think they would rather distract the reader than give additional good structured information. 

4) It is true that 300 minutes of playing video games are many, but the exposure is only one. Specify in the paper that the effects occur following a short-term exposure. The title could be changed to "The effects of prolonged single night session of videogaming on sleep and declarative memory"

Response: That is an excellent point. Thank you! We changed the title as suggested.

5) In the first phrase of the paragraph “Discussion of Results”, word “detrimental” is too strong. Replace with a more suitable term.

Response: We replaced the word with "negative".

6) Which Correct Words value is reported in the paper? Pre- or Post- sleep? This is unclear.

Response: We are sorry that this point is not clear. It depends on different parts in the paper. One of our main questions was if there is a difference in declarative memory between "Counter Strike" and "Monopoly" on the next morning, so this means post-sleep (results in l. 228-231)

In addition, we compared differences in declarative memory performance directly after the "Counter Strike"/ "Monopoly" stimulation, so this means pre-sleep (results in l. 246-252). We added "pre- and post-sleep" in some parts of the paper (l. 251; l. 341; l. 115; l. 231)

7) Why didn't you use a non-verbal declarative memory test? For example a visual declarative memory test. Other information could emerge (see Peracchia, S., & Curcio, G. (2018). Exposure to video games: effects on sleep and on post-sleep cognitive abilities. A sistematic review of experimental evidences)

Response: The most important point for the decision for the memory test was the possibility for repeating testing. The VLMT has got three comparable versions for every night. A non-verbal test could be a good alternative but it was a conscious decision to choose a verbal test because we did not want to use a screen at the board game night. 

8) Update the literature.

Response: We updated and checked the literature. Thank you for your suggestions.

Reviewer 2:

1) How was the VLMT administered? The text states that the list was memorized through five repetitions, but do not state whether words were presented one at a time, all together, etc., how much time participants were given to study the list, how much time passed in between repetitions, whether it was a computerized task, etc. Nor do the authors mention how much time participants were given for the free recall of the word list. It is not clear what is meant by “the recall was preannounced in each repetition” –does this mean participants were explicitly instructed to memorize the words?

Response: Sorry, this is not described clearly in the paper. Every participant knew that he would have to repeat the words.

The investigator read out all 15 words, then the participant had to reproduce all the memorized words. After that, the investigator read out all the words again and the participants repeated. In sum this happened five times. Afterwards a different list of 15 words was read out and the participant had to repeat them. Then they played Counter Strike or Monopoly. After the game (pre-sleep) there was another recall without a repetition of the words. On the next morning (post-sleep) the participants had to repeat the words again without a preceding repetition. There was no exact time limit for the recall but after approximately a minute the recall was stopped.

We added more information about the VLMT in the methods (l. 151-158).

2) For the 5 hours of gaming, were participants gaming continuously without breaks (other than the scheduled break at 8:55 to collect saliva)? It is also unclear whether the PCs were the sole source of light while playing video games or if there was additional ambient room lighting. Was the 20 lux of ambient lighting during the board game also measured at eye level? I am also curious how 20 lux was selected for the ambient lighting during the board game, as it is quite dim compared to ordinary room lighting. Finally, what lux was the room during the sleep period?

Response: There were no additional breaks during the stimulation. In fact, the participants were offered some food and drinks but they were told to play while eating.

There was additional room light while playing video games and the light intensity was measured at eye level in both conditions. The eye-screen-distance was about 30 cm. We chose a more ambient light during the board game because we wanted to reproduce a quiet board game evening at home like it may have typically happened 40 years ago. We added more information about that in the new version of the manuscript (l. 176; l. 373).

3) Were habitual bed/wake times taken into account during screening? Especially because sleep is one of the primary outcome measures, it is important to note whether the lab-scheduled sleep opportunities are at similar times to the participants’ habitual sleep times. If participants are attempting to sleep at times very different from their normal time, their sleep may be impacted independent of gaming condition.

Response: Unfortunately, there has been no adjustment of the personal habitual bed/wake times. Every night had a tight schedule in the sleep laboratory. The participants played the games for five hours together so it was not possible to create a timetable referring to everybody’s bed times. But you are definitely right. We added this topic to the discussion part (l. 326-330). 

4) How was the timing of the saliva samples chosen? Also, it would be helpful to report the average time and standard deviation of the fourth nocturnal saliva sample, so that readers are aware of the variability of this sample time.

Response: The aim of the saliva samples was to show the time course of the two hormones. The samples were taken before, in the middle and at the end of the game. To see the progression, the fourth samples was taken in the night. The extraction should cause the least possible disturbance so a time was chosen when less REM or slow wave sleep was expected (having in mind that the typical sleep cycle lasts 70-100 minutes). The mean and standard deviation of the fourth saliva sample time is 02.10 AM ± 19 minutes. We added this to the paper. The paper was not submitted with the actual version of table 3 and 4. We are sorry for that and changed it. In a former version, there was a little excel fault with regard to the sample times.

5) Why was the VigiMar test used to assess vigilance instead of the more commonly used Psychomotor Vigilance Test (PVT)? And given that the VigiMar is a very long test, were there any differences in time-on-task effects in the two conditions?

Response: The VigiMar was developed in Marburg where the study was conducted. The employees of the sleep laboratory know the test very well so it was easy to use it. In addition to that, we wanted to investigate a long-term vigilance test. The VigiMar is frequently used in Germany and described in a standard textbook of the German Sleep Society about methods in sleep medicine (Kompendium Schlafmedizin).

We haven´t included any data about the time-on-task effect into our data set yet so at this point we cannot compare this effect in the two conditions. Generally, it´s possible to adopt additional information into our digital data set but therefore another analysis of our analogue raw data set is necessary. In effect, it means that we have to go back to paper source data and to check it with our quality assurance as we´ve done with the rest of our data. For this process more time would be needed. Until now, we haven´t paid special attention to the time-on-task effect because we thought it would be rather slight but generally an analysis could be supplemented if a future revision is required.

6) The authors report significantly lower melatonin levels after video gaming compared to board games (pg 10) and state that this “indicates a phase delay of melatonin secretion”. While this clearly indicates melatonin suppression, and it is possible that evening video games could have caused a phase delay, the current experimental design does not allow them to assess whether there has actually been a phase shift. Similar on pg 14, line 360, the data show melatonin suppression but not necessarily a phase delay.

Response: You are right that the design does not allow to state a phase delay. We modified the wording accordingly (l. 272; l. 377).

7) Although the authors note that participants were scheduled to 7 hours time in bed, which is close to the average for German adolescents, they also report on page 5 that their participants reported sleeping on average 7.65 hrs in bed during the week and 9.02 hrs on the weekend. Thus, it is likely that they were either somewhat sleep restricting the participants, or at the very least scheduling wake times significantly different from the participants’ habitual wake. This should be mentioned in the discussion of scheduled time in bed.

Response: This is a limitation of the experimental design and has been added to the discussion of the methods (l. 326-330).

8) Minor typographical errors in the text:

a. Use consistent tense throughout the manuscript. E.g. on pg 2, line 28, change “does” to “did”; pg 14 line 355, change “seems” to “seemed”, etc.

b. Pg 5 in the ethics statement, it is not clear what “All participants were tired and informed…” is intended to say.

c. Pg 6, line 172 “brake” should be changed to “break”

d. Pg 15, line 393 “extend” should be changed to “extent”

Response: Thank you for your excellent and very thorough review. We have corrected the errors.

---

## [Decision Letter · Decision Letter 1]

24 Sep 2019

PONE-D-19-17611R1

The effects of prolonged single night session of videogaming on sleep and declarative memory

PLOS ONE

Dear Mrs. Hartmann,

Thank you for submitting your manuscript to PLOS ONE. After careful consideration, we feel that it has merit but does not fully meet PLOS ONE’s publication criteria as it currently stands. Therefore, we invite you to submit a revised version of the manuscript that addresses the points raised during the review process.

First of all I am Dr., not Mr.

Reviewers have re-evaluated the manuscript and Reviewer 2 identified further modifications to be included. 

Moreover, it had been asked according to journal requirements to add effect sizes, especially because the effect is deemed "surprisingly large" in conclusion. I was not able to find the effect sizes at the lines and tables Authors report in the response, but only means and p values. These should be added. I also suggest to modify those lines in conclusion; first, they seem unnecessary to comment on Authors' results, second they would be inappropriate unless the effect size is extremely high taking into consideration previous literature on the topic. 

We would appreciate receiving your revised manuscript by Nov 08 2019 11:59PM. To enhance the reproducibility of your results, we recommend that if applicable you deposit your laboratory protocols in protocols.io, where a protocol can be assigned its own identifier (DOI) such that it can be cited independently in the future. For instructions see: http://journals.plos.org/plosone/s/submission-guidelines#loc-laboratory-protocols

We look forward to receiving your revised manuscript.

Kind regards,

Stefano Triberti, Ph.D.

Academic Editor

PLOS ONE

Reviewers' comments:

Reviewer's Responses to Questions

**Comments to the Author**

1. If the authors have adequately addressed your comments raised in a previous round of review and you feel that this manuscript is now acceptable for publication, you may indicate that here to bypass the “Comments to the Author” section, enter your conflict of interest statement in the “Confidential to Editor” section, and submit your "Accept" recommendation.

Reviewer #1: All comments have been addressed

Reviewer #2: (No Response)

2. Is the manuscript technically sound, and do the data support the conclusions?

Reviewer #1: Yes

Reviewer #2: Yes

3. Has the statistical analysis been performed appropriately and rigorously? 

Reviewer #1: Yes

Reviewer #2: Yes

4. Have the authors made all data underlying the findings in their manuscript fully available?

Reviewer #1: Yes

Reviewer #2: Yes

5. Is the manuscript presented in an intelligible fashion and written in standard English?

Reviewer #1: Yes

Reviewer #2: Yes

6. Review Comments to the Author

Reviewer #1: Dear authors

I am very happy that you have made the suggested changes.

The study now appears clearer and more complete, suitable for publication.

On careful reading, the manuscript appears complete in all its parts.

It's technically valid and supports the conclusions.

Reviewer #2: I appreciate the opportunity to review the revised manuscript, and thank the authors for addressing the comments. My only remaining concern is that the revised description of the VLMT is still somewhat unclear. If the VLMT has been published previously, perhaps the authors could include a citation/reference with a more detailed description of the task? The parts I still found confusing are as follows:

Pg 6, L155 “…this list was read out by the investigator and reproduced by the participants for five times each”

It is unclear to me whether this means participants are verbally repeating the words as the investigator reads each word aloud, or if participants are being asked to repeat the full list from memory after the investigator has finished reading all 15 words, etc. From the comments to the reviewers, it sounds as though participants are being asked to freely recall all 15 words immediately after the investigator reads the full list, in which case it would be important to report whether participants received any feedback if they made errors or prompting if they were unable to successfully repeat all 15 words. The information about recall time should also be reported in the text.

Pg 6, L156-57 “…then another list with fifteen words was read to the participants who had to reproduce them”

Please clarify the purpose of this second list of words-- is it for masking/interference, or are participants tested on this list as well?

Again, I appreciate the authors’ work in addressing the reviewer comments in the discussion. I believe the manuscript is ready for publication with minor revisions to improve clarity.

7. PLOS authors have the option to publish the peer review history of their article (what does this mean?). If published, this will include your full peer review and any attached files.

Reviewer #1: Yes: Peracchia Sara

Reviewer #2: No

---

## [Author Response · Author response to Decision Letter 1]

18 Oct 2019

Dear Dr. Triberti, Dr. Peracchia and Reviewer 2,

Thank you for your reviews and the chance to revise the manuscript for a second time. First of all, please accept our sincere apologies for addressing you as Mr instead of the correct form Dr. In the following text we respond to your suggestions point-by-point again.

Dr. Stefano Triberti:

1) Moreover, it had been asked according to journal requirements to add effect sizes, especially because the effect is deemed "surprisingly large" in conclusion. I was not able to find the effect sizes at the lines and tables Authors report in the response, but only means and p values. These should be added. I also suggest to modify those lines in conclusion; first, they seem unnecessary to comment on Authors' results, second they would be inappropriate unless the effect size is extremely high taking into consideration previous literature on the topic.

Unfortunately, we have misunderstood your last suggestion for the first revision. You are right that no measures of effect sizes have been shown until now. We have calculated effect sizes for all statistically significant tests. We have added these to the tables and changed the text of the manuscript (i.e. l. 222-224, l. 235, l. 348-349, l. 412-414)

According to Cohen, all the reported r-values show a medium effect size for the statistically significant tests (r-values of 0.10, 0.30, and 0.50 to demarcate small, medium and large effects). Recent studies remark that Cohen's correlation guidelines are too exigent. They suggest to consider correlations of 0.10, 0.20 and 0.30 as relatively small, typical (medium) and relatively large for individual differences researchers. According to Gignac, our results show a large effect. (Gignac, Gilles E.; Szodorai, Eva T. (2016): Effect size guidelines for individual differences researchers. In: Personality and Individual Differences 102, S. 74–78. DOI: 10.1016/j.paid.2016.06.069.) 

In our manuscript we used the established and rather conservative evaluation of Cohen's guidelines. 

Dr. Sara Peracchia:

Thank you very much for your well composed review. We appreciate your positive decision.

Reviewer 2:

1) I appreciate the opportunity to review the revised manuscript, and thank the authors for addressing the comments. My only remaining concern is that the revised description of the VLMT is still somewhat unclear. If the VLMT has been published previously, perhaps the authors could include a citation/reference with a more detailed description of the task? The parts I still found confusing are as follows:

Pg 6, L155 “…this list was read out by the investigator and reproduced by the participants for five times each”

It is unclear to me whether this means participants are verbally repeating the words as the investigator reads each word aloud, or if participants are being asked to repeat the full list from memory after the investigator has finished reading all 15 words, etc. From the comments to the reviewers, it sounds as though participants are being asked to freely recall all 15 words immediately after the investigator reads the full list, in which case it would be important to report whether participants received any feedback if they made errors or prompting if they were unable to successfully repeat all 15 words. The information about recall time should also be reported in the text.

Pg 6, L156-57 “…then another list with fifteen words was read to the participants who had to reproduce them”

Please clarify the purpose of this second list of words-- is it for masking/interference, or are participants tested on this list as well?

We are really sorry that this part is still unclear. The participants were asked to recall the words after the investigator has finished reading all 15 words. The second list was just a distractor list. They were tested on this list as well, but only in one immediate recall which is not reported in the manuscript.

We have changed the description of the VLMT in the manuscript. Hopefully it is more understandable now:

“The VLMT consists of two lists, each with fifteen unrelated words (list A and list B). There is a parallel form for repeated testing. List A is read out by the investigator in five consecutive learning trials, each followed by an immediate free recall by the participants. This is followed by one round of reading and immediate recall of the distractor list. After the “Counter Strike: Global Offensive” resp. “Monopoly” stimulation (pre-sleep) and on Sunday morning (post-sleep) the participants were asked to recall the words from list A. There was no exact time limit for the recall, but it usually lasted less than one minute until subjects stated that they recalled no further words. There was no feedback about the number of correct recalled words.”

The VLMT represents a modified German version of the Rey Auditory Verbal Learning and Memory Test (RAVLT). 

A version of the test has been published by Witt for example (Witt, JA., Coras, R., Becker, A.J. et al. Brain Struct Funct (2019) 224: 1599. https://doi.org/10.1007/s00429-019-01857-1).

---

## [Editor Report · Decision Letter 2]

24 Oct 2019

The effects of prolonged single night session of videogaming on sleep and declarative memory

PONE-D-19-17611R2

Dear Dr. Hartmann,

We are pleased to inform you that your manuscript has been judged scientifically suitable for publication and will be formally accepted for publication once it complies with all outstanding technical requirements.

With kind regards,

Stefano Triberti, Ph.D.

Academic Editor

PLOS ONE
---

## [Editor Report · Acceptance letter]

5 Nov 2019

PONE-D-19-17611R2 

The effects of prolonged single night session of videogaming on sleep and declarative memory 

Dear Dr. Hartmann:

I am pleased to inform you that your manuscript has been deemed suitable for publication in PLOS ONE. Congratulations! Your manuscript is now with our production department. 

With kind regards,

on behalf of

Dr. Stefano Triberti 

Academic Editor

PLOS ONE